# Development of the early-life gut microbiome and associations with eczema in a prospective Chinese cohort

Man Kit Cheung,[1] Ting Fan Leung,[2,3] Wing Hung Tam,[3,4] Agnes S. Y. Leung,[2] Oi Man Chan,[2] Rita W. Y. Ng,[1,3] Jennifer W. K. Yau,[1] Lai-yuk Yuen,[4] Sylvia L. Y. Tong,[1] Wendy C. S. Ho,[1] Apple C. M. Yeung,[1] Zigui Chen,[1,3] Paul K. S. Chan[1,3]

**ABSTRACT** The first few years of life is a key period for the development of the gut microbiome. However, our current understanding of this topic is largely biased toward Western populations. In this study, we characterized the development and determinants of the gut microbiome in a prospective cohort of 112 term Chinese children by sequencing 713 stool samples collected at nine time points from birth to 3 years of age using 16S rRNA gene sequencing. We revealed alterations in the composition and alpha and beta diversities of the gut microbiota across the first 3 years of life. We identified mode of delivery, feeding mode, and intrapartum antibiotics as the major determinants of the early-life gut microbiome, the effects of all of which persisted up to 12 months. Importantly, by conducting a nested case–control study, we showed that alterations in the infant gut microbiota precede the development of eczema. Interestingly, we identified a depletion of *Bacteroides* and an enrichment of *Clostridium sensu stricto 1* in the gut microbiome of infants with eczema at 1 year old. The same patterns were also observed in C-section-born infants within the same time frames, suggesting a role of the gut microbiota in previously reported associations between C-section and increased risk of eczema. Our study has revealed important associations between the gut microbiome and eczema in infancy and has established the basis for potential prevention/treatment of eczema via modulation of the gut microbiota.

**IMPORTANCE** Eczema is a major allergic disease in children, which is particularly prevalent in Chinese children during their first year of life. In this study, we showed that alterations in the infant gut microbiota precede the development of eczema in a prospective Chinese cohort. In particular, we discovered enrichments of the genera *Clostridium sensu stricto 1* and *Finegoldia* in the cases at 3 and 1 month of age, respectively, which may represent potential targets for intervention to prevent eczema. Besides, we identified a depletion of *Bacteroides* from 1 to 6 months of age and an enrichment of *Clostridium sensu stricto 1* at 3 months in the eczema cases, patterns also observed in C-section-born infants within the same time frames, providing first evidence to support a role of the gut microbiota in previously reported associations between C-section and increased risk of eczema in infancy.

**KEYWORDS** atopic dermatitis, cesarean section, *Clostridium sensu stricto 1*, early childhood, gut microbiota, infancy, intrapartum antibiotics, longitudinal

The human gut microbiome in the first few years of life undergoes rapid and non-random development (1, 2). In particular, the first 1,000 days from conception to 2 years of age is a key period for the concurrent development of the gut microbiome and the immune system, which represents a "window of opportunity" to promote health via modulation of the gut microbiome through interventions (3). To date, several large prospective longitudinal studies have been carried out to characterize the early-life gut

Address correspondence to Paul K. S. Chan, paulkschan@cuhk.edu.hk, or Zigui Chen, ziguichen@cuhk.edu.hk.

Man Kit Cheung and Ting Fan Leung contributed equally to this article. Author order was determined by the amount of data analysis and writing.

The authors declare no conflict of interest.

See the funding table on p. 12.

10.1128/msystems.00521-23 **1**

microbiome; however, most of them are based on European and American populations (2, 4, 5). Since the infant gut microbiome is affected by ethnicity (6, 7), there is a need for similar studies in the other ethnic groups, especially in under-represented continents such as Southeast Asia and Africa (8). In fact, a few studies on the early-life gut microbiome in Asian populations have been published (7, 9, 10); however, these studies comprised a small sample size and/or had a limited sampling frequency (typically four time points).

The early-life gut microbiome is shaped by multiple maternal and postnatal factors (11). In particular, mode of delivery and breastfeeding are well known to have profound effects on the early-life gut microbiome (2, 12). However, the effects of other maternal and perinatal factors, such as parity and intrapartum antibiotics (13, 14), on the childhood gut microbiome are less clear, let alone their relative effects on other covariates and the period/duration these factors exert their effects. Meanwhile, the early-life gut microbiome is associated with the development of allergic diseases in children, including eczema, wheezing (asthma), and allergic sensitization (15). Among these allergic diseases, eczema affects up to 20% of children and represents one of the earliest manifestations of the atopic march (16). Therefore, it is important to identify the factors affecting eczema development, especially in Chinese children, who showed the highest incidence rate of eczema in the first year of life among other ethnic groups (17).

In this study, we established a prospective cohort of 112 Chinese children born at term in the Stool Microbiome and Allergic ReacTion (SMART) Baby Study and characterized their gut microbiome at nine time points from birth to 3 years of age by using 16S rRNA gene sequencing. Taking advantage of the allergy-related clinical data collected systematically and prospectively, we then conducted a longitudinal nested case–control study of eczema across the first year of life. Our specific objectives were to (i) characterize the development of the gut microbiome in a Chinese birth cohort during the first 3 years of life; (ii) determine the effects of mode of delivery, feeding mode, intrapartum antibiotics use, and 22 other maternal, perinatal, fetal, or environmental variables on the early-life gut microbiome; and (iii) identify alterations in the gut microbiome between infants with and without eczema.

## RESULTS

### Characteristics of the study cohort

A total of 120 newborns were initially recruited in the SMART baby cohort. Five of them (4.2%) withdrew from the study before providing the first (meconium) sample, and another three (2.5%) withdrew after providing the meconium sample, which failed to generate sufficient sequence reads for further analysis (see Materials and Methods). Eventually, a total of 112 newborns (93.3%) were included for analysis. These children were born between September 2017 and April 2018; 57 of them (50.9%) were male, and 85 (75.9%) were born vaginally (Table S1). The mean gestational age of birth and the mean maternal age were 39.3 (standard deviation [SD], 1.2) weeks and 32.5 (SD 4.0) years, respectively. Fifty-seven mothers of the newborns (50.9%) received intrapartum antibiotics. Twenty-four infants (21.4%) had been exclusively breastfed at 6 months old, and 64 (57.1%) were introduced solid food before 6 months. Forty-eight (42.9%) and 34 (30.4%) of the infants had physician-diagnosed eczema at the age of 6 and 12 months, respectively. The majority of these cases (>70%) were mild or moderate based on the scoring atopic dermatitis (SCORAD) and objective scoring atopic dermatitis (oSCORAD) scores. The mean age of eczema onset was 2.7 (SD 1.7) months. Twenty-six infants (23.2%) were positive for at least one tested allergen in skin prick test performed at 12 months old. The prevalence of other common allergic diseases at 12 months was low (wheezing, 0%; food allergy, 11.6%; and rhinitis, 2.7%).

## Composition and alpha and beta diversities of the gut microbiota change across the first 3 years of life

A total of 713 stool samples were collected from the study cohort at nine time points from birth to 3 years of age (Table S2). Most of the subjects (83/112, 74.1%) had samples from ≥6 time points. 16S rRNA V3–V4 gene sequencing of the stool samples generated a total of 8,577,863 quality-filtered sequence reads and an average of 12,014 (SD 6,532) reads per sample.

The meconium microbiota was dominated by the bacterial phyla Proteobacteria (40.6%) and Firmicutes (39.4%) (Fig. 1A). Starting from the first month of age, Actinobacteriota became the predominant bacterial phylum in the gut microbiota across the first year of life, with a peak relative abundance (55.7%) at 6 months. As the children grew, the relative abundance of Firmicutes and Bacteroidota increased and exceeded that of Actinobacteriota since 18 months, which then remained fairly stable up to 36 months. At the genus level, *Escherichia–Shigella* (35.8%) and *Enterococcus* (17.2%) dominated the meconium microbiota (Fig. 1B). From the first month onwards, *Bifidobacterium* and *Bacteroides* were the predominant genera of the gut microbiota across the first 3 years of life. As the children grew, the relative abundance of *Bifidobacterium* decreased, while that of *Bacteroides* increased until becoming relatively stable since 18 months. The relative abundance of *Faecalibacterium* increased gradually across the first 3 years of life starting from 6 months and became the third most abundant genus since 18 months. The relative abundance of *Blautia* also increased across the first 2 years of life since the first month. By contrast, the relative abundance of *Klebsiella*, *Clostridium sensu stricto 1*, and *Escherichia–Shigella*, which were abundant in the early months, decreased gradually along time. The relative abundance of the [*Ruminococcus*] *gnavus* group increased from 3 to 12 months and then dropped until 36 months.

Alpha diversity analysis showed that the richness, as measured by the observed number of amplicon sequence variants (ASVs), and Shannon diversity of the gut microbiota increased gradually across the first 3 years of life (Fig. 1C and D; Table S3). Principal coordinate analysis based on both unweighted and weighted UniFrac distances showed that the overall composition (beta diversity) of the gut microbiota also shifted along time ($P < 0.0001$) (Fig. 1E and F; Table S3). The biggest changes in the gut microbiome were observed between 6 and 12 months (Fig. 1; Table S3). These were accompanied by the enrichment of the genera [*Ruminococcus*] *gnavus* group, *Blautia*, and *Faecalibacterium* and the depletion of *Bifidobacterium*, *Escherichia–Shigella*, and *Enterococcus* (false discovery rate [FDR] < 0.1) (Table S4), as well as changes in several ASVs (Table S5), regardless of potential confounding factors including mode of delivery, feeding mode and intrapartum antibiotics use.

## Mode of delivery, feeding mode, and intrapartum antibiotics determine the gut microbiota composition in the first year of life

We examined the associations between the gut microbiota and 25 variables at each of the nine time points in the first 3 years of life. Results showed that the gut microbiota in the first year of life was mainly associated with mode of delivery, feeding mode and intrapartum antibiotics based on unweighted and weighted UniFrac distances (FDR <0.1) (Fig. 2; Fig. S1 and Table S2). Mode of delivery was the main determinant of the gut microbiota in the first month, explaining 6.3% and 11.9% of the total variance based on unweighted and weighted UniFrac distances, respectively, followed by intrapartum antibiotics use (unweighted UniFrac, 4.7%; weighted UniFrac, 6.5%). During the first year of life, the effect size of the two variables decreased gradually and was overtook by feeding mode since 6 months (Fig. 2A). Nonetheless, all these three variables remained significant contributors to the gut microbiota composition up to 12 months of age. Notably, feeding mode was only significant based on unweighted UniFrac distance, indicating that it mainly affects the low-abundance taxa. Other significant variables included prenatal probiotics use on the meconium microbiota, and the consumption of vitamins and maternal age on the gut microbiota at 18 and 30 months, respectively.

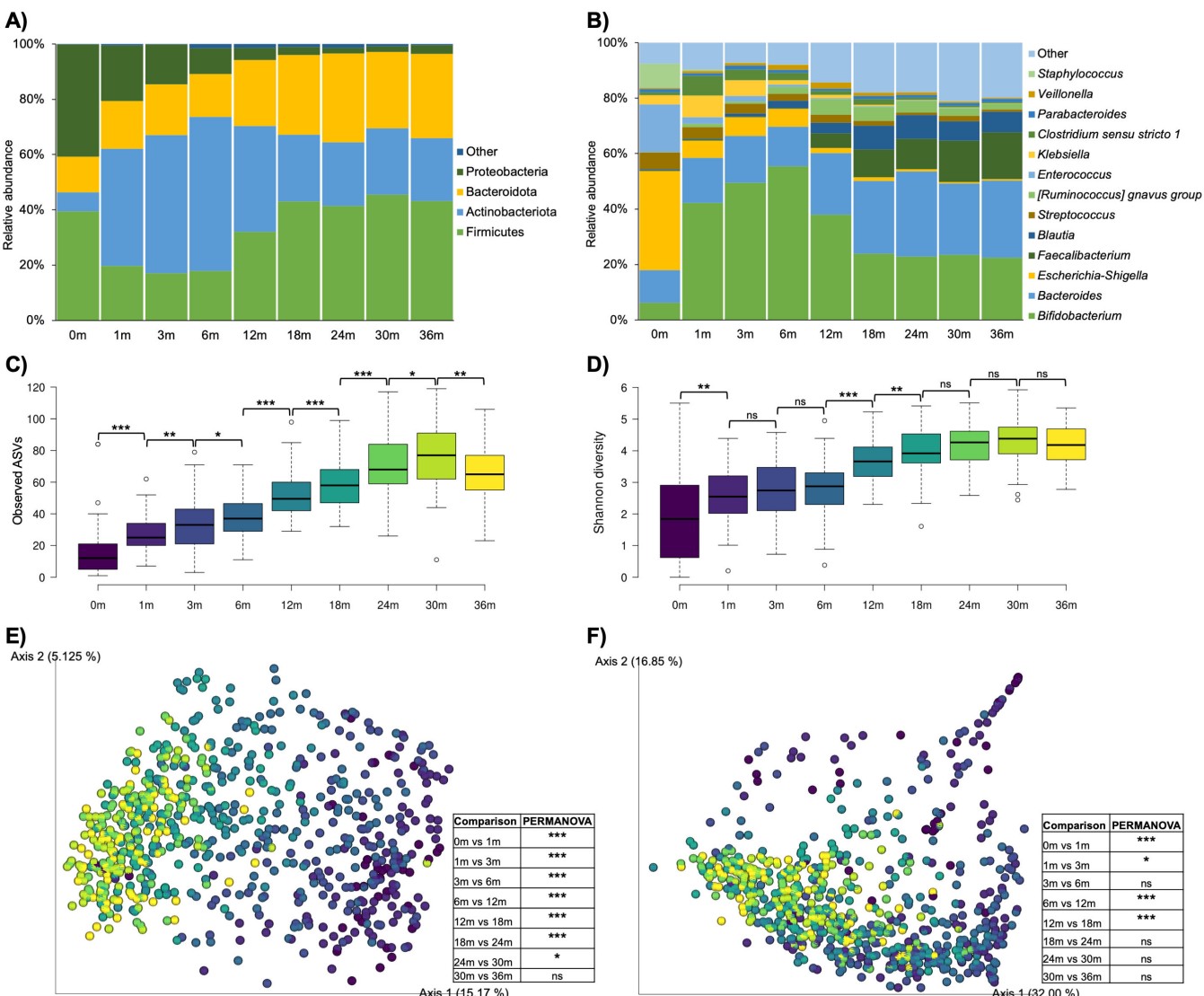

**FIG 1** Alterations in the taxonomic composition and alpha and beta diversities of the gut microbiota across the first 3 years of life. Taxonomic bar plots showing the major bacterial phyla (A) and genera (B) in the gut microbiota. Taxa with a mean relative abundance of <1% were grouped into "other." Box plots showing the alpha diversity of the gut microbiota as estimated by the number of observed ASVs (C) and Shannon diversity (D). Statistics between consecutive time points are shown. *$P < 0.05$, **$P < 0.01$, ***$P < 0.001$. Principal coordinate analysis plots showing the beta diversity of the gut microbiota based on unweighted (E) and weighted (F) UniFrac distances. Samples were colored according to the sampling time point following the color scheme used in panels C and D. *$P < 0.05$, **$P < 0.01$, ***$P < 0.001$. ASV, amplicon sequence variant; ns, not significant.

To understand which specific bacterial taxa of the gut microbiota were altered, we then performed differential abundance analysis on each of the three main significant variables at each time point in the first year of life while including the other two variables as potential confounders. Results showed that the gut microbiota of vaginally born infants was significantly enriched in the genera *Bacteroides* and *Parabacteroides* during the first 6 months of life when compared to infants born by C-section (FDR <0.1) (Fig. 3A). By contrast, five bacterial genera were significantly enriched in infants born by C-section, including *Clostridium sensu stricto 1* and *Enterobacter* at 1 and 3 months, *Enterococcus* and *Veillonella* at 1 month as well as *Klebsiella* at 3 and 6 months. The gut microbiota of exclusively breastfed infants was enriched in eight genera, including *Haemophilus* and *Staphylococcus* at 3 and 6 months, and *Limosilactobacillus*, *Bifidobacterium*, *Veillonella*, *Streptococcus*, *Lacticaseibacillus*, and *Megasphaera* at 12 months (Fig. 3B). By contrast, exclusively formula-fed infants were enriched in *Intestinibacter* at 3 and 6 months,

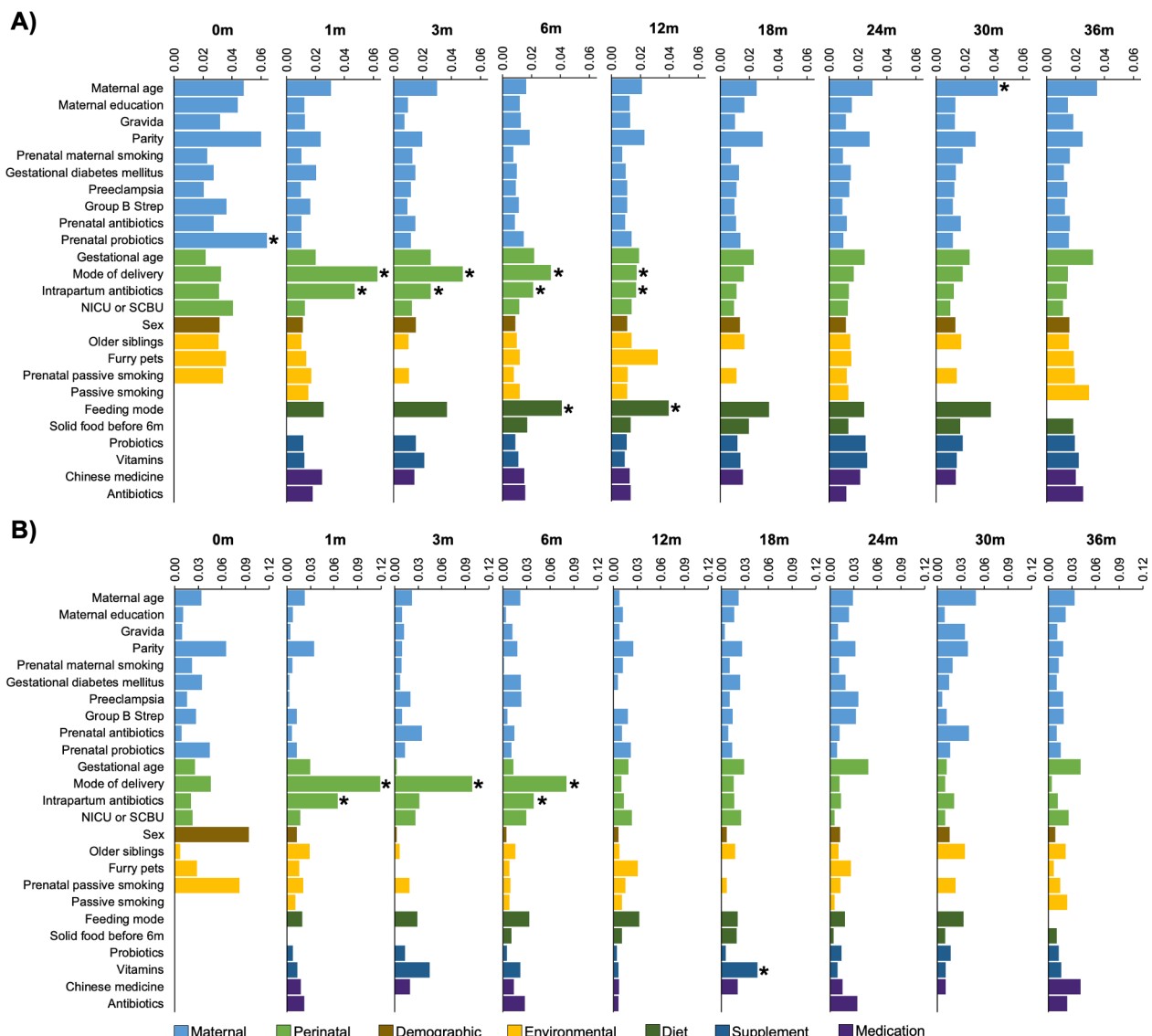

**FIG 2** Effect size ($R^2$) of 25 variables on the gut microbiota composition across the first 3 years of life as determined by permutational multivariate analysis of variance based on unweighted (A) and weighted (B) UniFrac distances. Bars were colored according to the categories of the variables. Effect size was not calculated when data were not collected or when the variable displayed limited variance. Groups within each variable are detailed in Table S2. *False discovery rate < 0.1. NICU, neonatal intensive care unit; SCBU, special care baby unit.

*Ruminococcaceae incertae sedis* at 6 months and *Flavonifractor* at 12 months. For subjects whose mothers received intrapartum antibiotics during labor, an enrichment of an unclassified genus of Enterobacteriaceae at 1 month and a depletion of *Eubacterium* at 12 months were observed (Fig. 3C). Differential abundance analysis performed at the ASV level revealed an enrichment of a *Clostridium sensu stricto 1* ASV in the gut microbiota of infants born by C-section at 3 and 6 months and an enrichment of an *Intestinibacter* ASV in exclusively formula-fed infants at 3 and 6 months, among others (Fig. S2).

## Alterations in the infant gut microbiota precede the onset of eczema

Taking advantage of the SMART baby cohort, we conducted a longitudinal nested case–control study of eczema. We built a subset (subset 1) containing 188 stool samples collected at four time points (1, 3, 6, and 12 months) from 23 infants with physician-diagnosed eczema at 12 months of age (cases) and 24 infants who never had eczema in their first year of life (controls). All basic characteristics, including mode of delivery,

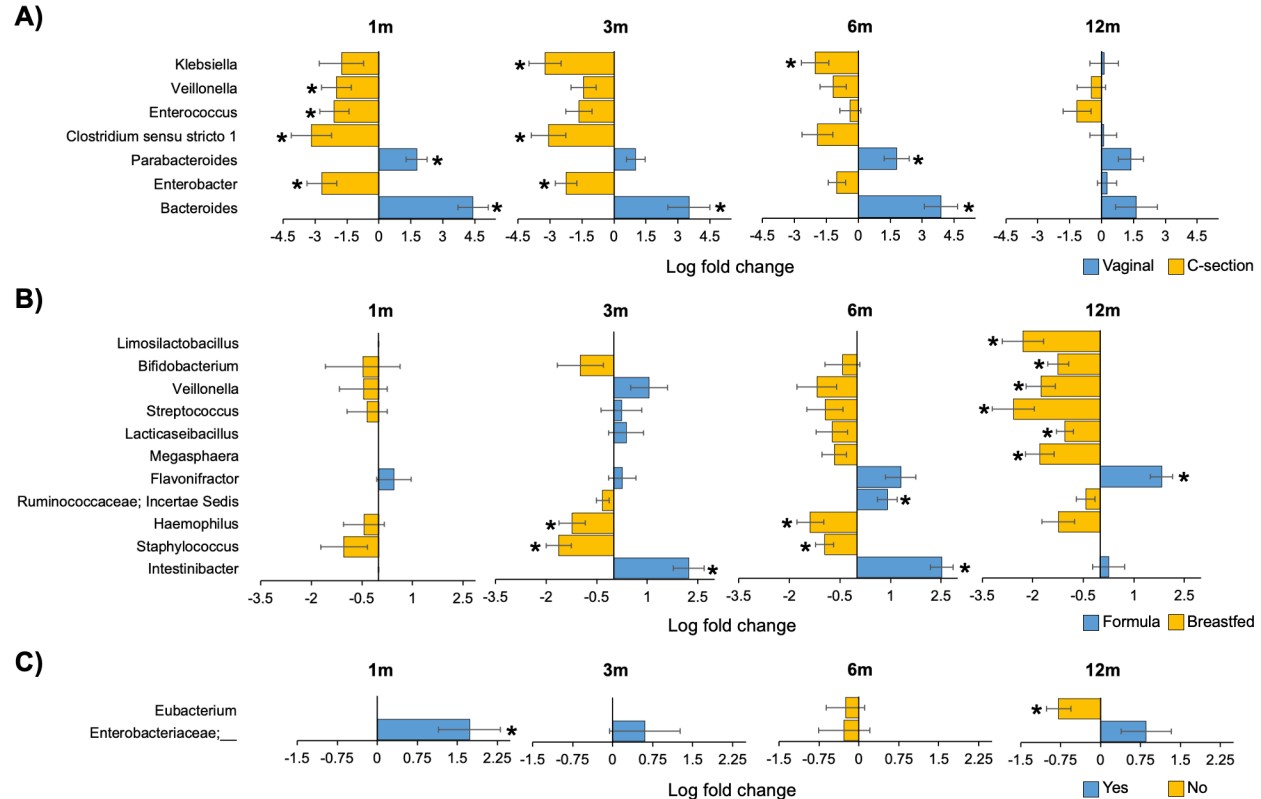

**FIG 3** Differentially abundant bacterial genera of the gut microbiota in the first year of life according to mode of delivery (A), feeding mode (B), and intrapartum antibiotic use (C). For calculations on a certain variable, the other two variables were included as covariates in the model. Length of bars represents the effect size (log fold change). Bars were colored according to the group in which a particular genus was enriched. Only genera showing significant difference at at least one time point were included here. Error bars are standard errors. *False discovery rate < 0.1.

feeding mode at all time points, and intrapartum antibiotics use, did not differ significantly between the two groups (*P* > 0.05) (Table S6).

No significant differences were observed in the alpha or beta diversity of the gut microbiota between the cases and controls at all time points (Table S7). However, differential abundance analysis revealed an enrichment of the genus *Clostridium sensu stricto 1* at 3 months (*P* < 0.01) and a depletion of *Bacteroides* at all four time points in the cases (*P* < 0.05), among other differentially abundant genera at specific time points (Fig. 4A). The mean relative abundance of *Bacteroides* in the controls was higher than the global mean of the full cohort from 1 to 6 months, whereas that in the cases was lower than the global mean at all time points except 6 months (Fig. 4B). For *Clostridium sensu stricto 1*, the mean relative abundance of the genus in both cases and controls was lower than the global mean in the first 6 months. Differences in the mean relative abundance were observed between the cases and controls as early as in the first month, albeit not statistically significant, which diminished over time. At the ASV level, a *Clostridium sensu stricto 1* ASV was found enriched in the cases from 1 to 6 months (*P* < 0.05) (Fig. S3). Intriguingly, this ASV was identical to the one depleted in vaginally born infants at 3 and 6 months (Fig. S2A).

Spearman's correlation analysis showed that the Shannon diversity at 12 months of age was negatively correlated to eczema severity as determined by oSCORAD score at the same time point (rho = −0.57, *P* = 0.017) (Fig. 4C). Besides, positive correlations were observed between the relative abundance of the genus *Clostridium sensu stricto 1* at 3 months and both SCORAD (rho = 0.61, *P* = 0.016) and oSCORAD scores (rho = 0.62, *P* = 0.013) at 6 months, whereas the relative abundance of *Bacteroides* at 6 months showed a

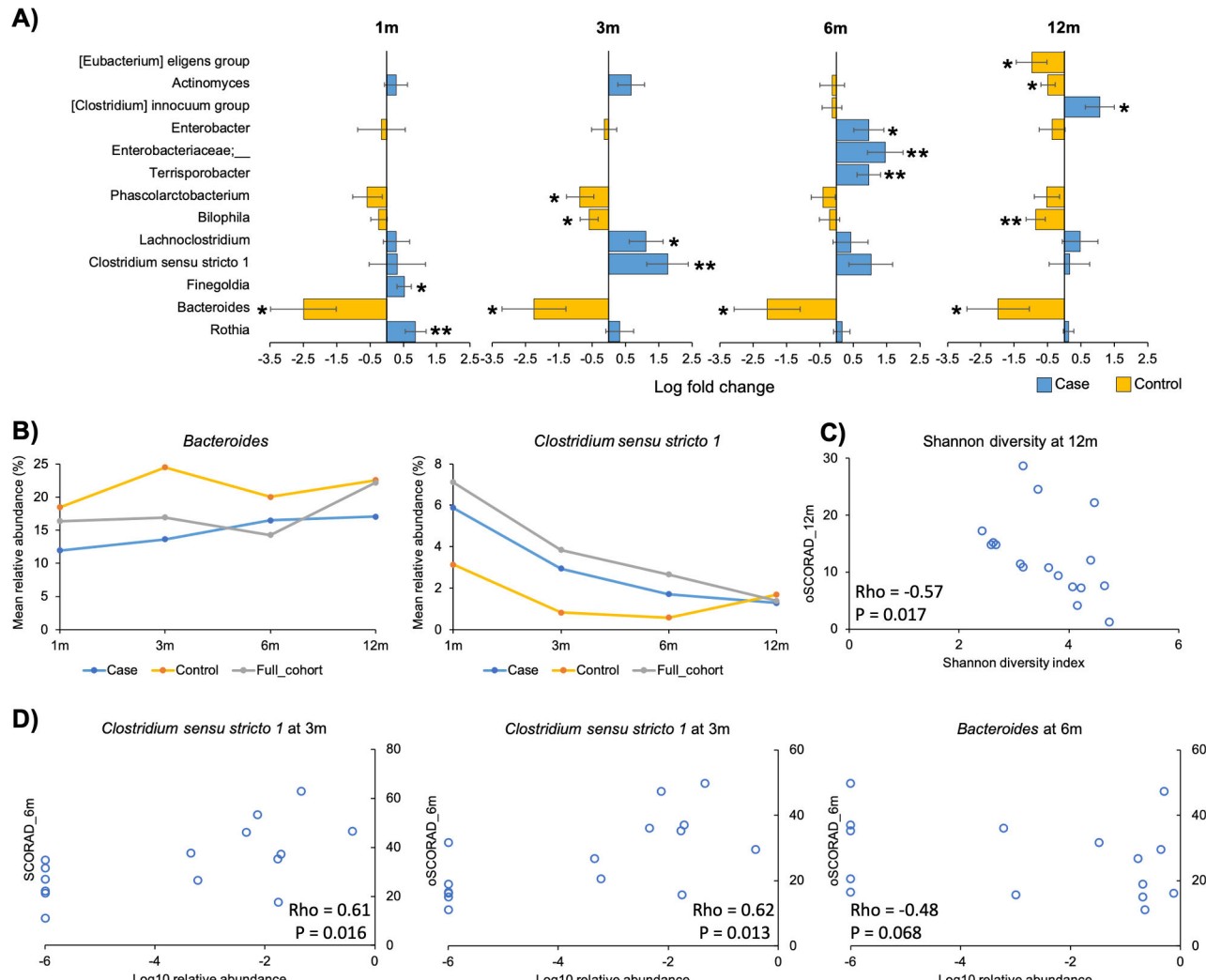

**FIG 4** Nested case–control study of the gut microbiome in eczema across the first year of life. (A) Differentially abundant bacterial genera of the gut microbiota in the first year of life between subjects with physician-diagnosed eczema at 12 months of age (cases) and those who never developed eczema by 12 months (controls). Length of bars represents the effect size (log fold change). Bars were colored according to the group in which a particular taxon was enriched. Only taxa showing significant difference at at least one time point were included here. Error bars are standard errors. *$P < 0.05$, **$P < 0.01$. (B) Trajectory of the mean relative abundance of the genera *Bacteroides* and *Clostridium sensu stricto 1* in the cases and controls during the first year of life. Data from the full cohort are also included as reference. (C) Spearman's correlation between Shannon diversity and oSCORAD score at 12 months of age. (D) Spearman's correlations between SCORAD/oSCORAD scores at 6 months of age and log10 relative abundance of *Clostridium sensu stricto 1* at 3 months and *Bacteroides* at 6 months. oSCORAD, objective Scoring Atopic Dermatitis.

trend to negatively correlate with the oSCORAD score at the same time point (rho = $-0.48$, $P = 0.068$) (Fig. 4D).

We then repeated the analysis on another subset (subset 2) comprising 189 stool samples collected at 1, 3, and 6 months from 33 infants with physician-diagnosed eczema at 6 months of age (cases) and 30 infants who never had eczema by the same age (controls). There were no significant differences in all basic characteristics between the cases and controls except maternal education ($P < 0.05$) (Table S6). Apart from a higher Shannon diversity observed in the cases at 6 months ($P < 0.05$), no significant differences were observed in the alpha or beta diversity of the gut microbiota between the cases and controls at all time points (Table S7). Differential abundance analysis revealed an enrichment of the genus *Clostridium sensu stricto 1* in the cases at 3 months ($P < 0.01$) (Fig. S4A). The mean relative abundance of this genus in the cases and controls was similar and lower than the global mean in the first month; however, it remained high

in the cases at 3 months, whereas that in the controls followed the global pattern and decreased (Fig. S4B). The same differentially abundant ASV of *Clostridium sensu stricto 1* in the analysis of subset 1 was also enriched in the cases from 1 to 3 months in the current analysis ($P < 0.05$) (Fig. S4C). No significant correlations were observed between eczema severity and the alpha diversity of the gut microbiota or the relative abundance of the genus *Clostridium sensu stricto 1* at all time points ($P > 0.05$) (data not shown). Given the small numbers of eczema cases at 24 ($n = 13$) and 36 ($n = 11$) months (Table S1), similar analysis was not performed at these time points.

## DISCUSSION

The obligate anaerobes *Bifidobacterium* and *Bacteroides* were the predominant genera of the gut microbiota in our cohort across the first 3 years of life. As the children grew, the relative abundance of *Bifidobacterium* decreased, while that of *Bacteroides* increased, a pattern also observed in Western populations (4, 18). Besides, the gradual increase in the bacterial richness and diversity of the gut microbiota across the first 3 years of life also agrees with other reports based on Western populations (2, 4). Collectively, these results suggest the presence of a common developmental programming in the early-life gut microbiome among ethnic groups, despite known effects of ethnicity on the baseline composition and diversity of the infant gut microbiome (6, 7).

At 36 months of age, the gut microbiota of our cohort was dominated by *Bacteroides* (27.8%), *Bifidobacterium* (22.5%), and *Faecalibacterium* (16.9%). This differs substantially from our previous results based on a healthy adult cohort from the same local population in which *Bacteroides*, *Bifidobacterium*, and *Faecalibacterium* had a relative abundance of 40.5%, 9.4%, and 2.9% in the gut microbiota, respectively (19), indicating that the gut microbiota at 3 years of age still differs from that of adults. This differs from the common belief that the adult-like gut microbiota configuration is reached by age 3 but supports a recent study which has shown that the gut microbiota has not yet reached adulthood complexity at 5 years of age (2, 11).

Our results confirm a major role of mode of delivery and feeding mode on shaping the gut microbiota in the first year of life (2, 12). The initially large but decreasing effect of mode of delivery and the initially small but increasing effect of feeding mode over time are consistent with other reports (2, 12, 20). Prior to FDR correction, the effect of mode of delivery remained significant up to 24 months. In fact, a previous study has reported a small but significant effect of mode of birth on the gut microbiota even at 5 years of age (2). These results suggest that mode of delivery may affect the gut microbiota of children for a period much longer than previously thought (4).

Apart from the mode of delivery and feeding mode, we showed that intrapartum antibiotics, a less-studied exposure, was also a major determinant of the gut microbiome in the first year of life. A few studies have shown that the effect of intrapartum antibiotics persists up to 12 months of age (14, 21, 22); however, none of them has reported data beyond 12 months. In the current study, by characterizing the gut microbiota up to 36 months of age, we confirm that intrapartum antibiotics can affect the infant gut microbiota up to 12 months and showed that, for the first time, the effect vanished thereafter.

Eczema is a major allergic disease in children, which is particularly prevalent in Chinese children during their first year of life when compared to other ethnic groups. Here, we conducted a nested case–control study of eczema in our Chinese cohort. Based on the results of the major subset (subset 1), we revealed no significant differences in the alpha or beta diversity of the gut microbiota between the cases and controls at all time points in the first year of life, in agreement with some other studies (23, 24). However, the lack of difference may be due to the fact that the cases in these studies mostly had mild or moderate disease. In fact, one study has reported a lower alpha diversity in the gut microbiota of moderate–severe, but not mild cases, at 6 months of age (25), suggesting a dose-dependent association between the severity of eczema and alpha diversity of the gut microbiota, at least at certain time points. This is supported by

a negative correlation between Shannon diversity and eczema severity (determined by oSCORAD) at 12 months of age observed in this study. In fact, a similar correlation has also been reported at 6 months of age (26).

Despite no significant changes in diversity were observed between the eczema cases and controls, the genus *Bacteroides* was significantly depleted in the cases at all four time points throughout the first year of life. The depletion of *Bacteroides* within the first year of life in infants with eczema is consistent with other studies (27, 28). Multiple species of *Bacteroides* are capable to educate and stimulate maturation of the host immune system via secretion of zwitterionic polysaccharides (29–31). The fact that this genus was enriched in the controls throughout infancy, instead of just at a single time point, strengthens the likelihood of its protective role against eczema development. The potential health benefit of *Bacteroides* on eczema is further supported by a negative trend observed between its relative abundance and eczema severity (determined by oSCORAD) at 6 months of age.

For the first time, we showed an enrichment of the genus *Clostridium sensu stricto 1* in the eczema cases at 3 months of age. In fact, a difference in the mean relative abundance was observed between the cases and controls as early as in the first month, albeit not statistically significant. Members of *Clostridium sensu stricto* are generally perceived as pathogenic (32). *Clostridium sensu stricto 1* may play a role in systemic immunity since its relative abundance is negatively associated with daily changes in neutrophil and monocyte counts (33). The potential deleterious effect of the genus on eczema development is supported by a positive correlation observed between its relative abundance at 3 months of age and eczema severity (determined by both SCORAD and oSCORAD) at 6 months. In fact, a previous study has also reported a positive correlation between *Clostridium sensu stricto* abundance at 3 months of age and eczema severity at 12 months (34).

Interestingly, the depletion of *Bacteroides* from 1 to 6 months of age and the enrichment of *Clostridium sensu stricto 1* at 3 months observed in the gut microbiota of the eczema cases aligned well with that observed in infants born by C-section within the same time frames. Previous research has shown that C-section is associated with a higher risk of eczema in infants within the first year of life (35). Besides, it has been suggested that alterations in the gut microbiota may represent an important link between mode of delivery and the increased prevalence of pediatric allergies following C-section birth (5). Therefore, our results have provided first evidence to support a role of the gut microbiota in the associations observed between C-section and increased risk of eczema in infancy. Of note, we did not observe an association between mode of delivery and eczema in our cohort, which could be masked by the small proportion of C-section (case: 2/23, 8.7%; control: 2/24, 8.3%).

Since the median onset time of eczema for the cases in the nested case–control study was at 3 months of age (interquartile range: 2–4 months), the alterations in the gut microbiota in the first month were particularly intriguing since they precede eczema onset and thus suggest a causal relationship with the disease. Apart from the depletion of *Bacteroides*, we discovered a significant enrichment of the low-abundance (mean < 0.1%) genus *Finegoldia* in the gut microbiota of the cases in the first month. *Finegoldia magna*, the sole species within *Finegoldia*, is capable to weaken the host immune defense by secreting the superantigen protein L (36). Superantigens bind to antigen-presenting cells and T cells, allowing them to interact without the constraint of an antigenic peptide and resulting in excessive production of T-cell cytokines (37). In fact, the role of superantigens secreted by *Staphylococcus aureus*, frequently isolated from the skin of eczema patients, on eczema is well studied (37). Our results suggest that *Finegoldia magna* in the gut microbiota may promote eczema via a similar mechanism.

Collectively, our results indicate that alterations in the infant gut microbiota precede the development of eczema. Besides, we have identified a list of bacteria that may represent potential microbial markers for early prognosis or diagnosis of eczema. If the potential roles of these bacteria in eczema development can be confirmed in animal

models or future prospective studies, interventions may be conceived to prevent the onset and/or attenuate the severity of later developed eczema via modulating the abundance of these bacteria, especially during the critical window in the first few months of age.

Strengths of this study include its prospective design, which has greatly minimized recall bias. Longitudinal sampling at multiple time points from birth up to 3 years of life has also allowed detailed characterization of the early-life gut microbiome. Besides, collection of a multiplicity of maternal and perinatal metadata has allowed investigations of the absolute and relative effects of some less-studied exposures on the gut microbiota across the first few years of life. Lastly, eczema status and severity were determined by pediatricians and using validated scores rather than self-report by parents. Nevertheless, there are also limitations to our study. First, 16S rRNA sequence data used here have hindered inference of species/strain-level taxonomy and do not provide functional insights. Second, the relatively small numbers of cases and controls in the nested case–control study of eczema have resulted in a limited statistical power for correction of multiple comparisons. Lastly, most of the eczema cases had mild or moderate disease, which may have underestimated the effect if there was a dose-dependent association.

## Conclusions

To conclude, by characterizing the longitudinal development of the early-life gut microbiome in a prospective cohort of Chinese children born at term, we showed that the alpha diversity of the gut microbiome increased and the beta diversity altered across the first 3 years of life—patterns also observed in Western populations—suggesting the presence of a concerted developmental programming of the early-life gut microbiome irrespective of ethnicity. Besides, we showed that apart from mode of delivery and feeding mode, intrapartum antibiotic use was also a main determinant of the early-life gut microbiome, the effect of which persisted up to 12 months of age but vanished afterward. Importantly, by conducting a nested case–control study of eczema, we revealed that alterations in the infant gut microbiota precede the development of eczema, identified a list of bacteria that may represent potential microbial markers for early prognosis or diagnosis of eczema, and provided first evidence to explain previously reported associations between C-section and increased risk of eczema. Further studies using animal models are needed to confirm the role of these taxa in eczema development.

## MATERIALS AND METHODS

### Study population

The SMART baby cohort comprised term newborns in Hong Kong recruited before delivery in the antenatal ward from 2017 to 2018. The inclusion criteria were ethnic Chinese and willing to provide follow-up stool samples throughout the first 3 years of life. The exclusion criteria were preterms and those not staying in Hong Kong for most of the time. Newborns enrolled were followed prospectively from birth to 3 years of age with home visit scheduled in the first month and subsequent clinic visits at 6, 12, 24, and 36 months. Stool samples were collected at the first defecation (meconium), 1, 3, and 6 months after birth, then every 6 months until 3 years of age (total of nine time points). Meconium samples were collected at the postnatal ward, whereas follow-up stool samples, approximately peanut sized, were collected by the parents at home with sterile spoons into 1.5 mL of transport medium (Norgen Biotek) following instructions provided and mailed back to the laboratory. Samples were stored at −80°C before DNA extraction.

Maternal and perinatal information obtained from the computerized Clinical Management System of the Hong Kong Hospital Authority included maternal age, gravida, parity, gestational diabetes mellitus, preeclampsia, group B *Streptococcus*,

gestational age, mode of delivery, intrapartum antibiotics use, the newborn's sex, 1- and 5-min Apgar scores, admission to the neonatal intensive care unit or the special care baby unit, and birth weight. Other information was provided by the mother at the antenatal ward and during home visit in the first month after delivery, and subsequently via standardized questionnaires during clinic visits. This included maternal education, prenatal probiotics and antibiotics use, smoking habit of mother and family members, parental history of allergies, number of older siblings, furry pets, as well as use of antibiotics and Chinese medicine. Information about feeding mode, introduction of solid food, as well as probiotic and vitamin use were collected by both questionnaire and dietary record. Eczema was diagnosed in follow-up assessments at 6, 12, 24, and 36 months by pediatricians during clinic visits based on the widely adopted Hanifin and Rajka criteria, and severity of eczema was evaluated using both SCORAD and oSCORAD. Allergic sensitization was assessed at 12 months during clinic visit by skin prick test with *Dermatophagoides pteronyssinus*, egg, cow's milk, peanut, wheat, soy bean, and mixed fish. Wheezing, food allergy and rhinitis at 12 months was obtained from the questionnaire.

## 16S rRNA gene sequencing

DNA was extracted from the stool samples using DNeasy PowerSoil Kit (QIAGEN) following the manufacturer's instructions. The V3–V4 region of the 16S rRNA gene was then amplified using universal primers 341F (5′-CCT ACG GGN GGC WGC AG-3′) and 806RB (5′-GGA CTA CNV GGG TWT CTA AT-3′). PCR products were pooled and sequenced on a MiSeq instrument (Illumina) following the 2 × 300 bp paired-end sequencing protocol. Negative controls (distilled water as template), positive controls (ZymoBIO-MICS Microbial Community DNA Standard [Zymo Research]), and technical replicates (randomly selected DNA samples) were also amplified and sequenced for quality control.

## Microbiome analysis

Microbiome analysis was performed with QIIME2 v.2020.11 as previously described unless otherwise specified (38, 39). In brief, paired-end reads were quality filtered, joined, and denoised to generated amplicon sequence variants (ASVs) using q2-dada2 (40). Taxonomy was assigned using Naïve Bayes classifier trained on the V3–V4 region of the SILVA 138 SSU Ref NR 99 data set (41). Mitochondrial, chloroplast, and phylum-unclassified reads were discarded. Samples with <3,000 quality-filtered sequence reads were also removed. Alpha and beta diversity analyses were performed using q2-diversity after rarefying the samples to the smallest number of reads. Alpha diversity metrics computed included the number of observed ASVs and Shannon diversity, whereas unweighted and weighted UniFrac distances were used for beta diversity estimation.

## Statistical analysis

Differences in alpha diversity between time points or groups were tested using Kruskal–Wallis test with the alpha-group-significance function in q2-diversity, whereas differences in beta diversity were tested using permutational multivariate analysis of variance based on 9,999 permutations with the beta-group-significance function in q2-diversity. For each time point, the univariate effect size of each metadata variable was calculated using the adonis function in q2-diversity with 9,999 permutations after removal of samples with missing or uncertain data. *P* values were adjusted using the Benjamini–Hochberg procedure to control for multiple comparisons. Differentially abundant genera and ASVs were identified using the compositionality-aware q2-ANCOMBC in QIIME2 v.2022.2 while controlling for potential confounding covariates (42). For the nested case–control study, categorical variables were compared between case and control groups using Fisher's exact test or chi-squared test, whereas continuous variables were compared using Mann–Whitney test. Spearman's correlation was used to test for associations between the relative abundance of specific bacterial

genera and SCORAD scores. Differences were considered statistically significant when the P value was < 0.05 or the false discovery rate was <0.1.

## ACKNOWLEDGMENTS

We thank the parents and children for taking part in this study. We thank Nancy Li, Nancy Cheng, Amy Chang, Cecily Leung, Yehao Chen, Yuping Song, and Liz Li for their assistance in this study. We also thank the Core Utilities of Cancer Genomics and Pathobiology at the Department of Anatomical and Cellular Pathology of the Chinese University of Hong Kong for the service of 16S rRNA gene sequencing.

The study was supported by a seed fund for gut microbiota research by the Faculty of Medicine, The Chinese University of Hong Kong, Hong Kong Special Administrative Region, People's Republic of China.

M.K.C. analyzed the data and wrote the manuscript. T.F.L., W.H.T., A.S.Y.L., O.M.C., and L.-Y.Y. collected clinical information. J.W.K.Y., S.L.Y.T., A.C.M.Y., and W.C.S.H. collected samples and performed the laboratory analysis. J.W.K.Y. was involved in data analysis. R.W.Y.N. assisted in the data interpretation. Z.C. provided insights into the data analysis. P.K.S.C. conceived and supervised the study and acquired funding. T.F.L., W.H.T., and P.K.S.C. critically reviewed the manuscript.

The authors report no conflict of interest.

## AUTHOR AFFILIATIONS

[1]Department of Microbiology, Faculty of Medicine, The Chinese University of Hong Kong, Hong Kong Special Administrative Region, Hong Kong, China
[2]Department of Paediatrics, Faculty of Medicine, The Chinese University of Hong Kong, Hong Kong Special Administrative Region, Hong Kong, China
[3]Centre for Gut Microbiota Research, The Chinese University of Hong Kong, Hong Kong Special Administrative Region, Hong Kong, China
[4]Department of Obstetrics and Gynaecology, The Chinese University of Hong Kong, Hong Kong Special Administrative Region, Hong Kong, China

## AUTHOR ORCIDs

Man Kit Cheung  http://orcid.org/0000-0002-2764-2113
Paul K. S. Chan  http://orcid.org/0000-0002-6360-4608

## FUNDING

| Funder | Grant(s) | Author(s) |
| --- | --- | --- |
| Faculty of Medicine, The Chinese University of Hong Kong | | Paul K. S. Chan |

## DATA AVAILABILITY

All sequence data generated from this study were deposited in the NCBI Sequence Read Archive under BioProject accession PRJNA912769. A STORMS checklist is available at Zenodo.

## ETHICS APPROVAL

The Stool Microbiome and Allergic Reaction baby study was approved by the Joint Chinese University of Hong Kong, New Territories East Cluster Clinical Research Ethics Committee (CREC Ref. No.: 2016.637). All mothers provided written informed consent to participate in this study.

## ADDITIONAL FILES

The following material is available online.

## Supplemental Material

**Supplemental figures (mSystems00521-23-s0001.pdf).** Figures S1 to S4.
**Supplemental tables (mSystems00521-23-s0002.xlsx).** Tables S1 to S7.

## Open Peer Review

**PEER REVIEW HISTORY (review-history.pdf).** An accounting of the reviewer comments and feedback.

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
