## [Reviewer comments · mSystems]

Development of the Early-life Gut Microbiome and Associations with Eczema in a Prospective Chinese Cohort

Man Kit Cheung, Ting Fan Leung, Wing Hung Tam, Agnes Sze Yin Leung, Oi Man Chan, Rita Ng, Jennifer Wing Ki Yau, Lai-yuk Yuen, Sylvia Tong, Wendy Ching-sze Ho, Apple Yeung, Zigui Chen, and Paul Chan

Corresponding Author(s): Paul Chan, Chinese University of Hong Kong

Review Timeline:

Submission Date:

May 22, 2023

Accepted:

July 20, 2023

Editor: Emily Cope

Reviewer(s): Disclosure of reviewer identity is with reference to reviewer comments included in decision letter(s). The following individuals involved in review of your submission have agreed to reveal their identity: Loo Wee Chia (Reviewer #2)

Transaction Report:

DOI: <https://doi.org/10.1128/msystems.00521-23>

July 20, 2023

Prof. Paul K. S. Chan
Chinese University of Hong Kong
Department of Microbiology
Prince of Wales Hospital
Hong Kong
Hong Kong

Re: mSystems00521-23 (Development of the Early-life Gut Microbiome and Associations with Eczema in a Prospective Chinese Cohort)

Dear Prof. Paul K. S. Chan:

Your manuscript has been accepted, and I am forwarding it to the ASM Journals Department for publication. For your reference, ASM Journals' address is given below. Before it can be scheduled for publication, your manuscript will be checked by the mSystems production staff to make sure that all elements meet the technical requirements for publication. They will contact you if anything needs to be revised before copyediting and production can begin. Otherwise, you will be notified when your proofs are ready to be viewed.

If you would like to submit a potential Featured Image, please email a file and a short legend to mSystems@asmusa.org. Please note that we can only consider images that (i) the authors created or own and (ii) have not been previously published. By submitting, you agree that the image can be used under the same terms as the published article. File requirements: square dimensions (4" x 4"), 300 dpi resolution, RGB colorspace, TIF file format.

We recognize that the video files can become quite large, and so to avoid quality loss ASM suggests sending the video file via <https://www.wetransfer.com/>. When you have a final version of the video and the still ready to share, please send it to mSystems staff at mSystems@asmusa.org.

Sincerely,

Emily Cope
Editor, mSystems

Journals Department
Dear Editor,

The manuscript "**Development of the Early-life Gut Microbiome and Associations with Eczema in a Prospective Chinese Cohort**" by Cheung et al. investigated association between early life gut microbiota and manifestation of eczema in Chinese population. The authors have provided a thorough response to the comments from the previous review and have addressed my major concerns.